# A New Indicator to Differentiate Thyroid Follicular Inclusions in Cervical Lymph Nodes from Patients with Thyroid Cancer

**DOI:** 10.3390/ijms24010490

**Published:** 2022-12-28

**Authors:** Chieko Otsubo, Zhanna Mussazhanova, Hirokazu Kurohama, Gulzira Shalgimbayeva, Nozomi Ueki, Yuki Matsuoka, Madina Madiyeva, Shinya Sato, Hiroyuki Yamashita, Masahiro Nakashima

**Affiliations:** 1Department of Tumor and Diagnostic Pathology, Atomic Bomb Disease Institute, Nagasaki University Graduate School of Biomedical Sciences, Nagasaki 852-8523, Japan; 2Department of Fundamental Medicine, Al-Farabi Kazakh National University, Almaty 050040, Kazakhstan; 3Department of Diagnostic Pathology, Nagasaki University Hospital, Nagasaki 852-8523, Japan; 4Department of Pathophysiology, Semey Medical University, Semey 071400, Kazakhstan; 5Nagasaki Regional Supportive Center for Diagnostic Pathology, Nagasaki University Hospital, Nagasaki 852-8523, Japan; 6Department of Radiology, Semey Medical University, Semey 071400, Kazakhstan; 7Yamashita Thyroid Hospital, Fukuoka 812-0034, Japan

**Keywords:** thyroid follicular inclusion, ectopic thyroid tissue, nodal metastasis, 53BP1, immunofluorescence

## Abstract

Nodal metastasis is crucial for determining the stage of well-differentiated thyroid cancer (WTC) in patients older than 55. Well-formed thyroid follicular inclusions (TFIs) are occasionally encountered in the cervical lymph nodes (LNs) of patients with papillary thyroid carcinoma (PTC), and it is difficult to determine whether they are true nodal metastases or ectopic thyroid tissues (ETT). This study aimed to elucidate the impact of the expression of the DNA damage response molecule TP53-binding protein 1 (53BP1) using immunofluorescence (IF) as a biomarker to differentiate TFIs in cervical LN by comparing the mutation analyses of primary thyroid cancers. The data demonstrated the necessity for the differential diagnosis of true metastases from ETT among TFIs in cervical LNs. PTC-like nuclear features using hematoxylin–eosin staining combined with immunohistochemistry for conventional biomarkers of PTC, including BRAF^V600E^ protein, were most helpful in identifying metastatic follicular-patterned carcinomas. In conclusion, IF analysis of 53BP1 expression could be an excellent ancillary technique to distinguish metastatic carcinoma or ETT from TFIs in LNs, particularly in cases other than *BRAF^V600E^*-mutated PTC.

## 1. Introduction

The status of nodal metastasis is crucial in determining the stage of well-differentiated thyroid cancer (WTC) in patients older than 55. Well-formed thyroid follicular inclusions (TFIs) are occasionally encountered in cervical lymph nodes (LNs) resected from patients with WTC [1,2], and it is difficult to determine whether they are true nodal metastases or ectopic thyroid tissues (ETTs) if they exhibit few papillary thyroid carcinomas (PTC)-like nuclear features (NFs). ETTs in cervical LN are reportedly found in 4.7 to 10% of autopsy cases without thyroid cancer [1,2,3], suggesting a nature other than metastatic carcinoma, yet aberrantly localized tissue is found in nodes during ontogeny. Thus, we are required to distinguish ETT from nodal metastasis in TFIs found in resected cervical LNs from patients with thyroid cancer.

Several genetic alterations, such as *BRAF*, *RET/PTC* rearrangements, *RAS*, *RET*, *TERT*-promoter, and *CTNNB*, are known to be involved in thyroid carcinogenesis [4,5,6,7,8,9]. Among them, point mutations in *BRAF^V600E^* are most frequently observed in PTCs, especially in adult patients [10]. Recently, Lin et al. demonstrated that immunohistochemistry (IHC) for the mutant BRAF^V600E^ protein (mBRAF) is useful for identifying metastatic follicular-patterned carcinoma from PTC harboring mBRAF among TFIs by comparing the results of PCR analysis for the *BRAF^V600E^* gene [2]. However, because IHC for mBRAF cannot identify metastatic carcinomas carrying other mutations, such as *NRAS codon 61* mutations, *RET/PTC* rearrangements, and *NTRK3-ETV6* translocations, other ancillary techniques are required to more appropriately discriminate metastasis from ETTs in cervical LN resected from patients with thyroid cancer.

TP53 binding protein-1 (53BP1) is a DNA damage-response (DDR) molecule that rapidly localizes at sites of DNA double-strand breaks (DSBs), forming nuclear foci to activate the downstream repair process [11,12,13,14]. Using immunofluorescence (IF), we previously demonstrated that the number of 53BP1 nuclear foci increases in the thyroid gland of rats after radiation in a dose-dependent manner [15,16]. Furthermore, our dual-color IF analysis detected frequent co-localization of 53BP1 and γH2AX nuclear foci in human follicular thyroid carcinomas (FTCs) and irradiated thyroid glands of rats, suggesting that the endogenous activation of the DDR pathway in cancer cells is a hallmark of genomic instability [16,17]. Therefore, this study aimed to elucidate the impact of 53BP1 expression using IF as a biomarker to differentiate TFIs in cervical LNs by comparing mutation analyses of primary thyroid cancers.

## 2. Results

### 2.1. Presence of BRAF^V600E^ Mutation, 53BP1, mBRAF, HBME-1, and Galectin-3 Expression in Primary Thyroid Tumors and Metastatic PTC in Cervical LNs

The results of the present study are shown in Table 1, with the clinicopathological profiles of the patients and representative images of the primary thyroid tumors and TFIs shown in Figure 1 (Appendix A) and Figure 2 (Appendix A), respectively. The *BRAF^V600E^* mutation was detected in 22 (95.7%) of the 23 cases of primary PTC and not in the other primary tumors (FTC, WDT-UMP, and FA). No *NRAS codon 61* mutations were detected in any of the primary tumors. All cases of primary PTC possessing the *BRAF^V600E^* mutation and their corresponding metastatic PTC were positive for all three PTC biomarkers, that is, mBRAF, HBME-1, and galectin-3, and also for abnormal 53BP1 expression. However, one case of *BRAF^V600E^* mutation-negative primary PTC was negative for mBRAF, yet positive for HBME-1, galectin-3, and abnormal 53BP1 expression. Among other types of primary tumors, two cases of FTC and WDT-UMP were positive for abnormal 53BP1 expression and negative for other PTC biomarkers. FA did not show any immunoreactivity in the IHC analysis. Furthermore, three cases of PTC and their corresponding metastatic PTC, which were employed as negative controls for mBRAF and positive controls for other biomarkers in IHC, were positive for HBME-1, galectin-3, and abnormal 53BP1 expression and negative for mBRAF expression. 

### 2.2. Expression of 53BP1, mBRAF, HBME-1, and Galectin-3 in TFIs of Cervical LNs

In TFIs without PTC-like NFs, immunoreactivity for PTC-associated biomarkers, such as mBRAF, HBME-1, and galectin-3, was only observed in two FTC cases that showed 53BP1 expression and primary FTC. Conversely, in TFIs with questionable PTC-like NFs, six of the nine cases (66.7%) were positive for all three PTC biomarkers, abnormal 53BP1 expression, metastatic PTC, and *BRAF^V600E^*-mutated primary PTC. The remaining three cases of TFIs with questionable PTC-like NFs were negative for mBRAF, although their primary PTC was *BRAF^V600E^*-mutated. Figure 3 shows a clear difference in 53BP1 expression using IF between the TFIs with and without PTC-like NFs in one field.

## 3. Materials and Methods

### 3.1. Sample Collections

This study defined TFI as follows: (1) it consisted of only well-formed follicular-patterned thyroid tissue in cervical LNs that were resected from patients with primary thyroid tumors; (2) PTC-like NFs were determined by following the scoring system previously published [18]—scores 0 and 1 were considered as the absence of PTC-like NFs, score 2 was considered as questionable PTC-like NFs, and score 3 was considered as metastatic PTC; (3) cases with psammoma body/microcalcification and desmoplastic change in the stroma were excluded; and (4) lesions that were too small to obtain available sections any further were excluded. We retrospectively detected 23 cases (1.5%) of TFI from 1504 cases of PTC with nodal metastases that were surgically resected at the Yamashita Thyroid Hospital in Fukuoka, Japan, from January 2018 to August 2021. One case (3.7%) of TFI was found in twenty-seven cases of well-differentiated tumor-uncertain malignant potency (WDT-UMP), and one case (0.4%) of TFI was also found in two hundred and forty-eight cases of follicular adenoma (FA). Additionally, 2 cases (1.9%) of TFI were found in 107 cases of FTC that were surgically resected at Nagasaki University Hospital, Japan, from January 2000 to January 2019. Thus, a total of 27 cases of TFI in cervical LNs were included in this study (Figure 4). Three cases of PTC with nodal metastases possessing mutations other than *BRAF^V600E^*, such as *SQSTM1-NTRK3*, *ETV6-NTRK3*, and *HRASQ61R*, were also available as negative controls for mBRAF and positive controls for other conventional biomarkers in IHC.

All available samples were formalin-fixed and paraffin-embedded (FFPE) tissues. The final diagnosis of all cases was histologically confirmed at the Department of Tumor and Diagnostic Pathology, Nagasaki University, according to the diagnostic criteria of the WHO Classification of Tumors of Endocrine Organs (4th edition) [19]. Representative pathological images of TFIs in this study are shown in Figure 5.

### 3.2. IF Analysis for 53BP1 Expression

The nuclear expression of 53BP1 was examined using dual-color IF analysis with cytokeratin expression to assess 53BP1 expression in epithelial cells in nodal tissues. Following deparaffinization and antigen retrieval via microwave treatment in Target Retrieval Solution, pH 6.0 (Agilent Technologies, Santa Clara, CA, USA), tissue sections of 4 µm thickness were preincubated with Dako Protein Block, Serum-Free (DakoCytomation, Glostrup, Denmark). For dual-color IF, sections were incubated with anti-53BP1 rabbit polyclonal antibody (1:1000; A200-272A; Bethyl Labs, Montgomery, TX, USA) and anti-pan-cytokeratin mouse monoclonal antibody (prediluted; AE1/AE3 ab961; Abcam, Cambridge, UK) for 1 h at 24 °C. The slides were subsequently incubated with Alexa Fluor 488-conjugated goat anti-rabbit (Molecular Probes Inc., Eugene, OR, USA) and Alexa Fluor 594 F(ab’)-conjugated goat anti-mouse antibodies phenylindole dihydrochloride (Vysis Inc., Downers Grove, IL, USA), analyzed, and photographed using a high-standard fluorescence microscope (Biorevo BZ-X710; KEYENCE Japan, Osaka, Japan). All 53BP1 nuclear expression signals were analyzed in each case at 1000-fold magnification. The expression of 53BP1 was classified into three types based on the number of nuclear foci: (1) stable type, faint nuclear staining and no or one nuclear focus; (2) DDR type, two or more discrete nuclear foci; and (3) diffuse type, intense heterogeneous nuclear staining. In this study, the DDR and diffuse types were considered positive for abnormal 53BP1 expression.

### 3.3. IHC for mBRAF, Hector Battifora Mesothelial Epitope (HBME)-1, and Galectin-3 Expression

The semi-serial sections used in IF analysis were also analyzed using IHC to detect mBRAF and other conventional biomarkers to estimate the malignant potential of thyroid tumors, such as HBME-1 and galectin-3 [20,21]. Following deparaffinization and antigen retrieval by heating tissue sections in an autoclave for 20 min in Cell Conditioning Buffer 1 (Ventana Medical Systems, Inc., Tucson, AZ, USA) for mBRAF and in citrate buffer (pH 6.0) for HBME-1 and galectin-3, the sections were immersed in 0.3% H_2_O_2_ solution for 5 min to block the endogenous peroxidase activity. Thereafter, sections were incubated with anti-mBRAF (mutated V600E) mouse monoclonal antibody (1:100; VE1 ab228461; Abcam), anti-HMBE-1 mouse monoclonal antibody (1:100; A200-272A; Dako, TX, USA), and anti-galectin-3 mouse monoclonal antibody (pre-diluted; API-3174; Biocare Medical, Pacheco, CA, USA) for 1 h at 24 °C with primary antibodies in a humidified chamber. The immunoreactivity of these proteins in each case was developed using diaminobenzidine peroxidase substrate and evaluated at 400-fold magnification. The level of immunoreactivity was categorized into four groups according to the percentage of positive cells as follows: (1) negative, 0%–5%; (2) low, 5%–30%; (3) moderate, 30%–60%; and (4) high, ≥60%. In this study, both moderate and high immunoreactivity levels were considered as staining-positive. These stainings were validated with *BRAF^V600E^*-positive PTC, epithelioid mesothelioma, and colon cancer FFPE tissues as a positive control of IHC for mBRAF, HBME-1, and galectin-3, respectively, and *BRAF^V600E^*-negative PTC and normal thyroid FFPE tissues as a negative control of IHC for mBRAF and HBME-1/galectin-3, respectively.

### 3.4. DNA Extraction

To analyze the thyroid cancer-related mutations in primary tumors, the genomic DNA of the tumor in each case was extracted from FFPE tissues via macrodissection with a guide slide stained with hematoxylin and eosin. Each FFPE section of 10 µm-thickness was dewaxed with 80% xylene in a tube. The dewaxed sample was washed twice with absolute ethanol and centrifuged at 15,000× *g* for 15 min at 24 °C. Following drying, the samples were digested for 12 h with proteinase K at 56 °C. DNA extraction was performed using a Maxwell RSC DNA FFPE Kit (Promega, Madison, WI, USA) and Maxwell RSC Instrument (Promega) according to the manufacturer’s protocol. The concentration of double-strand DNA was quantified with a Nanodrop 1000 spectrophotometer (Thermo Fisher Scientific, Wilmington, NC, USA).

### 3.5. Droplet Digital PCR (ddPCR) Analysis for BRAF^V600E^ Mutations

Mutation-specific primer/probe combinations were used to detect *BRAF^V600E^* mutations according to the manufacturer’s protocol (catalog # 12001037; Bio-Rad, Hercules, CA, USA). Amplifications were performed in a 20 μL reaction mixture containing ddPCR Supermix (Bio-Rad), 900 nM primers, 250 nM probe, and 5 μL (for BRAF assay) DNA. Droplets were generated using an automatic droplet generator QX200 AutoDG (catalog #1864002; Bio-Rad) and analyzed with a QX200 Droplet Reader (catalog #1864003; Bio-Rad). The analysis of ddPCR was performed with a QX200 ddPCR system according to the manufacturer’s instructions (Bio-Rad). The data were analyzed using QuantaSoft^TM^ software (catalog #1864011; Bio-Rad). The negative control contained nuclease-free water, whereas the positive control contained DNA from the FFPE samples.

### 3.6. NRAS Codon 61 Mutations by ddPCR Analysis

*NRAS codon 61* mutations were analyzed by ddPCR with an *NRAS Q61* Screening Kit (catalog #12001006; Bio-Rad), which detects five mutations in *NRAS codon 61* (Q61K, Q61L, Q61R, Q61H 183A > T, and Q61H 183A > C), according to the manufacturer’s protocol. A total of 20 µL of each reaction mixture containing 1.5 µL of extracted DNA was loaded in a sample well of a DG8 Cartridge (catalog #1864008; Bio-Rad), followed by the addition of 60 µL of Droplet-Generation Oil for Probes (catalog #1863005; Bio-Rad) in oil wells. Droplets were generated using a QX200 Droplet Generator (catalog #1864002; Bio-Rad) and transferred to a clean 96-well plate. The plate was sealed with the PX1 PCR Plate Sealer (catalog #1814000; Bio-Rad) and PCR was subsequently performed on a C1000 Touch Thermal Cycler (catalog #1851197; Bio-Rad). Following PCR amplification, each droplet was analyzed using a QX200 Droplet Reader (catalog #1864003; Bio-Rad) and QuantaSoft^TM^ software (catalog #1864011; Bio-Rad).

## 4. Discussion

The pathological consideration of TFIs in cervical LNs resected from patients with PTC is controversial. Some experts may always consider them metastatic thyroid carcinoma, whereas others may consider them benign ETTs when they lack the nuclear features of PTC. ETT consists of well-formed colloid-filled follicles and is suggested to represent a failure in the proper migration of thyroid follicles, which can be found anywhere along the midline, from the foramen cecum to the neck [22]. Lin et al. demonstrated the availabilities of IHC for the expression of mBRAF, HBME-1, and galectin-3, and mutation analyses for the *BRAF^V600E^* allele and *NRAS/KRAS* mutations to identify ETT; the accuracy of IHC for mBRAF expression to identify ETT had 89% sensitivity and 100% specificity in comparison with mutation analyses [2]. Their results suggest the necessity of histological discrimination for true metastasis from ETT among TFIs in cervical LNs resected from patients with PTC.

IHC for conventional biomarkers for PTC, such as mBRAF, HBME-1, and galectin-3, may help identify ETT in resected LNs. In addition to conventional biomarkers, this study demonstrated a clear difference in 53BP1 expression using IF between TFIs without PTC-like NF and with questionable PTC-like NFs or metastatic PTC. The IHC results for conventional markers and 53BP1 expression in metastatic PTC were identical to those in primary PTC, whereas those in TFIs without PTC-like NFs were negative. This suggested that TFIs without PTC-like NFs were not true metastases, but ETT in LNs as well as in LNs from cases of WDT-UMP and FA.

Notably, the IHC results for conventional PTC-related biomarkers cannot exclude metastatic FTC or metastatic PTC possessing the wild-type *BRAF* gene. Indeed, the TFIs without PTC-like NFs in LNs from FTC cases showed no immunoreactivity for these biomarkers. However, because of abnormal 53BP1 expression and primary FTC, they were suggested to be metastatic FTC in LNs. Furthermore, our method successfully demonstrated abnormal 53BP1 expression in mBRAF-negative PTC cases harboring mutations other than *BRAF^V600E^*. We have already reported that the number of 53BP1 nuclear foci detected using IF increases with higher biological aggressiveness in FFPE tissues of thyroid follicular tumors, such as FA, minimally invasive follicular carcinoma (FC), and widely invasive FC [16]. A recent study using liquid-based cytology samples obtained from resected thyroid follicular tumors suggested that the frequency of abnormal 53BP1 expression could be an attractive candidate biomarker to distinguish FC from FA [17]. Thus, IF analysis of 53BP1 expression may be a good ancillary technique to distinguish metastatic carcinoma or ETT from TFIs in LNs.

Our previous experiments based on IF analysis using FFPE tissues also found that the number of 53BP1 nuclear foci increased in irradiated rat thyroid glands in a dose-dependent manner, and frequent co-localization of 53BP1 and γH2AX nuclear foci occurred in human FTCs, suggesting the endogenous activation of DDR pathways in cancer cells as a hallmark of genomic instability [16,17]. Additionally, we showed that abnormal 53BP1 expression is closely associated with carcinogenesis in several organs [23,24,25,26,27,28,29,30,31,32]. Thus, analyzing 53BP1 expression using IF can be useful for estimating the malignant potential of human tumors. For instance, diffuse patterns were significantly associated with high-grade urothelial carcinoma with chromosomal instability, as demonstrated by the multicolor fluorescence in situ hybridization and poor prognosis [27]. We hypothesize that the pattern of 53BP1 expression can be an indicator of genomic instability in tumor cells. As it is technically possible to automate the quantification of DDR type by using single-cell imaging, our method may become a universal tool to estimate the malignant potential of tumors with liquid-based cytology from not only the thyroid gland, but also the urinary tract. Moreover, a defect in 53BP1 induces DNA damage checkpoint defects, impaired DNA repair, and hypersensitivity to ionizing radiation [33,34]. Thus, our method for estimating the impaired 53BP1 functions may have implications in estimating the sensitivity to ionizing radiation or chemotherapy in carcinomas. Further study is required to clarify the therapeutic implications of the type of 53BP1 expression in human tumors.

In summary, this study demonstrated the necessity of differentially diagnosing true metastases from ETT among TFIs in cervical LNs. In the diagnosis setting, PTC-like NF using hematoxylin–eosin staining combined with IHC for conventional biomarkers of PTC, including mBRAF, is most useful for identifying metastatic follicular-patterned carcinomas. Furthermore, we suggest that IF analysis of 53BP1 expression may be a good ancillary technique to distinguish metastatic carcinoma or ETT from TFIs in LNs, particularly in cases other than *BRAF^V600E^*-mutated PTC.

## Figures and Tables

**Figure 1 ijms-24-00490-f001:**
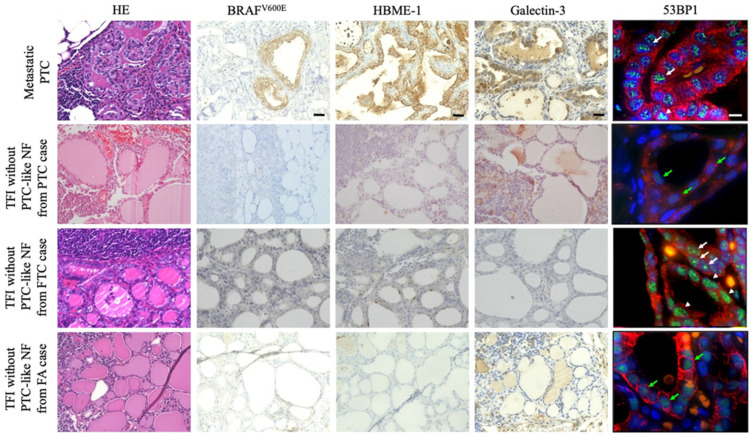
Immunohistochemistry (IHC) of biomarkers for papillary thyroid carcinoma (PTC), such as *BRAF^V600E^*, Hector Battifora mesothelial epitope (HBME)-1, and galectin-3 expression, and double immunofluorescence (IF) for p53-binding protein 1 (53BP1) (green) and pan-cytokeratin (AE1/AE3) (red) expression to detect follicular epithelium in primary thyroid tumors. In IF, the arrowheads and white and green arrows indicate the diffuse, DDR, and stable types of 53BP1 expression, respectively. The scale bars represent 30 µm and 10 µm in HE/IHC and IF, respectively.

**Figure 2 ijms-24-00490-f002:**
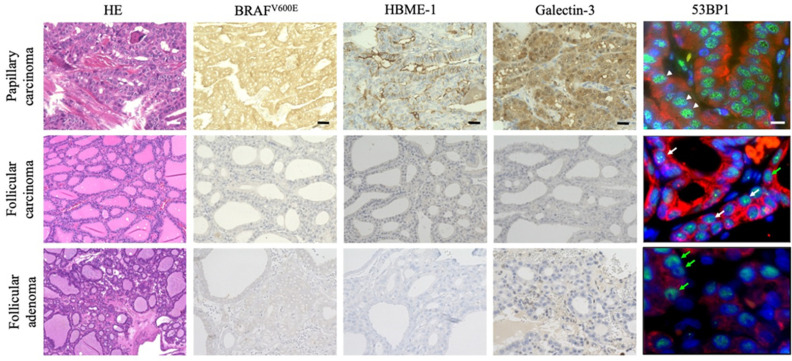
Immunohistochemistry (IHC) of biomarkers for papillary thyroid carcinoma (PTC), such as *BRAF^V600E^*, Hector Battifora mesothelial epitope (HBME)-1, and galectin-3 expression, and double immunofluorescence (IF) for p53-binding protein 1 (53BP1) (green) and pan-cytokeratin (AE1/AE3) (red) expression to detect follicular epithelia in metastatic PTC and thyroid follicular inclusions (TFIs) in cervical lymph nodes. All three biomarkers for PTC were positive in metastatic PTC and not in other TFIs without PTC-like nuclear features (NFs). In IF, the arrowheads and white and green arrows indicate the diffuse, DDR, and stable types of 53BP1 expression, respectively. The scale bars indicate 30 µm and 10 µm in HE/IHC and IF, respectively.

**Figure 3 ijms-24-00490-f003:**
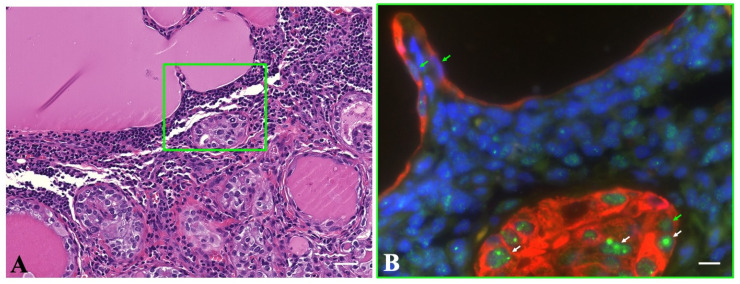
Double immunofluorescence for p53-binding protein 1 (53BP1) (green) and pan-cytokeratin (AE1/AE3) (red) expression to detect follicular epithelium in a cervical lymph node from a papillary thyroid carcinoma (PTC) case. It exhibits both metastatic PTC (lower part) and thyroid follicular inclusion (TFI) without PTC-like nuclear features (NFs) (upper part) (**A**). In (**B**), surrounded by a green square in (**A**), a metastatic PTC shows the DNA damage response type with two or more discrete nuclear foci, whereas the TFI without PTC-like NFs shows a stable type and lymphocytes in the background. In (**B**), the white and green arrows indicate the DDR and stable types of 53BP1 expression, respectively. The scale bars indicate 30 µm and 10 µm in (**A**,**B**), respectively.

**Figure 4 ijms-24-00490-f004:**
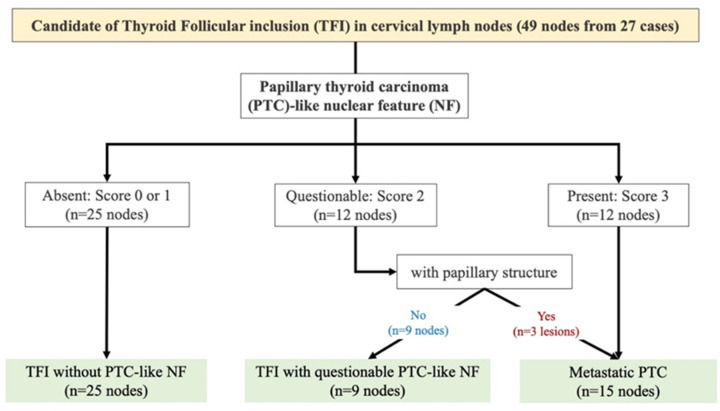
The number of thyroid follicular inclusions (TFIs) in cervical lymph nodes.

**Figure 5 ijms-24-00490-f005:**
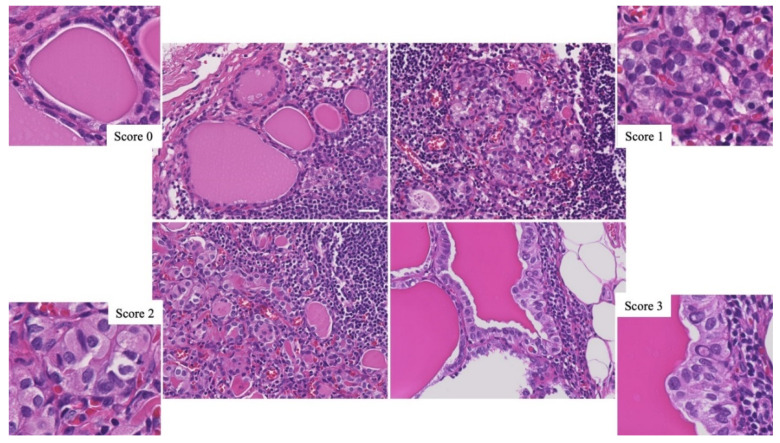
Representative pathological images of thyroid follicular inclusions (TFIs) in cervical lymph nodes. The scale bar indicates 30 µm.

**Table 1 ijms-24-00490-t001:** Summary of clinicopathological profiles of the patients and results in this study.

F/M	Age	pT	Type	*BRAF^V600E^ (Others)*	Primary Thyroid Tumor				Metastatic PTC				TFI without PTC-Like NF				TFI with Questionable PTC-Like NF			
					mBRAF	HBME	Gal	53BP1	mBRAF	HBME	Gal	53BP1	mBRAF	HBME	Gal	53BP1	mBRAF	HBME	Gal	53BP1
F	55	1a	PTC	+	+	+	+	NA	+	+	+	+	−	−	−	−	+	+	+	+
M	65	1a	PTC	+	+	+	+	+	+	+	+	+	−	−	−	−	+	+	+	+
F	52	1b	PTC	+	+	+	+	+	+	+	+	+	−	−	−	−	−	−	−	−
M	28	1b	PTC	+	+	+	+	+	+	+	+	+	−	−	−	−	+	+	+	+
F	30	3a	PTC	+	+	+	+	+	+	+	+	+	−	−	−	−	+	+	+	+
F	60	1a	PTC	+	+	+	+	+	+	+	+	+	NA	NA	NA	−	+	+	+	+
F	80	1a	PTC	+	+	+	+	+	+	+	+	+	−	−	−	−				
M	45	1b	PTC	+	+	+	+	+	+	+	+	+	−	−	−	NA				
F	43	1b	PTC	+	+	+	+	+	+	+	+	+	−	−	NA	−				
F	48	1b	PTC	+	+	+	+	+	+	+	+	+	−	−	−	−				
M	61	1b	PTC	+	+	+	+	+	+	+	+	+	−	−	−	−				
F	46	2	PTC	+	+	+	+	+	+	+	+	+	−	NA	−	−				
F	69	4a	PTC	+	+	+	+	+	+	+	+	+	−	−	−	−				
F	57	1b	PTC	+	+	+	+	+					−	−	−	−	−	−	−	−
F	35	1b	PTC	+	+	+	+	+					−	NA	NA	NA	−	NA	NA	NA
F	58	3b	PTC	+	+	+	+	+	+	+	+	+					+	+	+	+
F	72	1a	PTC	+	+	+	NA	+					−	−	NA	−				
F	65	1a	PTC	+	+	+	+	+					−	NA	−	−				
F	31	1a	PTC	+	+	+	+	+					−	−	−	−				
F	64	1b	PTC	+	+	+	+	+					−	NA	−	−				
F	30	1b	PTC	+	+	+	+	+					−	−	−	−				
F	56	2	PTC	+	+	+	+	+					−	−	−	−				
F	65	1a	PTC	− (Unkown)	−	+	+	+	-	+	+	+								
M	28	2	FTC	− (Unkown)	−	−	−	+					−	−	−	+				
M	83	3b	FTC	− (Unkown)	−	−	−	+					−	−	−	+				
F	63		WDT-UMP	− (Unkown)	−	−	−	+					−	−	−	-				
F	46		FA	− (Unkown)	−	−	−	−					−	−	−	-				
F	38	1b	PTC	− (*SQSTM1-NTRK3*)	−	+	+	+	−	+	+	+								
F	48	2	PTC	− (*ETV6-NTRK3*)	−	+	+	+	−	+	+	+								
F	62	3b	PTC	− (*HRASQ61R*)	−	+	+	+	−	NA	NA	NA								

pT, T-factor by UICC classification; TFI, thyroid follicular inclusion; PTC, papillary thyroid carcinoma; NF, nuclear features; LN, lymph node; HBME, Hector Battifola mesothelial cell-1; Gal, Galectin-3; 53BP1, TP53-binding protein 1; WDT-UMP, well differentiated tumor of uncertain malignant potential; FTC, follicular thyroid carcinoma; FA, follicular adenoma; NA, not available; Blue, PTC cases as a negative control for mBRAF.

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
