# Peer review of "A New Indicator to Differentiate Thyroid Follicular Inclusions in Cervical Lymph Nodes from Patients with Thyroid Cancer"

_ijms, 2022, doi:10.3390/ijms24010490_

Round 1
Reviewer 1 Report
Read the paper with interest, but as a surgeon, despite my strong interest in thyroic cancer, I do not feel adequate to review this paper that is otherwise a pathology paper. I would recomend to seek the opintion of pathologist.
Author Response
Read the paper with interest, but as a surgeon, despite my strong interest in thyroic cancer, I do not feel adequate to review this paper that is otherwise a pathology paper. I would recomend to seek the opintion of pathologist.
Response: Thank you for indication about this.
Reviewer 2 Report
Comments on IJMS 2006515, entitled “A New Indicator to Differentiate Thyroid Follicular Inclusions in Cervical Lymph Nodes from Patients with Thyroid Cancer” by Otsubo et al.
The manuscript is an interesting investigation on biomarkers to distinguish ectopic thyroid inclusions on lymph nodes from thyroid cancer metastasis. However, there are some changes that must be amended to increase comprehensibility.
MAJOR COMPULSORY CHANGES
The manuscript should be reviewed by an English native person, as there are multiple spelling lapses.
MATERIAL AND METHODS
There is not a clear cut between “material and methods” and “results”. In fact, table 1 (Summary of the results) is mentioned in material and methods section…
2.1. Sample collections- Authors should have a control group of people with ETT devoid thyroid tumours (for example ETT collected from cadaver’s devoid thyroid tumors).
2.3. IHC for mBRAF, Hector Battifora mesothelial epitope (HBME)-1, and galectin-3 expression- Please insert information regarding positive and negative controls used on IF and in IHQ.
DISCUSSION SECTION
Lines 160-161 - Please clarify the sentence, as it is difficult to understand “This suggested not true metastasis; however, it suggested ETT in LNs and in LNs from cases of WDT-UMP and FA.” as there are some verbs missing.
Lines 169-170 - Please clarify the sentence: “IF increases with higher malignant potential in FFPE tissues of thyroid follicular tumors, such as FA, minimally invasive follicular carcinoma (FC), and widely invasive FC (5).” as FA (follicular adenoma) has not malignant potential. I suggest authors to replace “malignant potential” to “biological aggressiveness”.
REFERENCES
There are very few references (n=12), and most of the references cited (7/12) have one or more authors from the present manuscript. More references should be insert in the manuscript.
MINOR CHANGES
The text has different font letters; font must be uniformed.
INTRODUCTION
Line 9- Amend the sentence “of autopsy cases are without thyroid cancer” to “of autopsy cases without thyroid cancer”
MATERIALS AND METHODS SECTION
2.1. Sample collections lines 4-5 - Please replace “…following the scoring system in a previous paper…” to “…following a scoring system previously published…” or …following the scoring system described by Nikiforov et al. (2016).”
2.1. Sample collections lines 24-25- Before this sentence “…Representative pathological images of TFI in this study are shown in Figure 2….” there should be a paragraph. And the previous sentence should be insert after the reference to “formalin-fixed, paraffin embedded tissues”, as classification is always performed after the routine histologic procedure. I suggest the following sequence: “All available samples were formalin-fixed and paraffin-embedded …. Representative pathological images… Clinicopathological profiles of the patients are summarized in Table 1.”
2.2. IF analysis for 53BP1 expression, line 7- Please replace “…using a high standard all-in-one fluorescence microscope…” to “…using a high standard fluorescence microscope…
2.2. IF analysis for 53BP1 expression, line 10- Please replace “…nuclear foci:1” for “…nuclear foci: 1”.
RESULT SECTION
3.1., lines 75-76- Please remove from the title “expression using IF and immunoreactivity of” as it was already mentioned in the manuscript
3.2., lines 120-121- Please remove from the title “using IF and immunoreactivity of” as it was already mentioned in the manuscript
3.2., lines 128-130- The sentence “These data indicated the presence of benign 128 ETT in cervical LNs resected from patients with PTC, which should be differentiated from 129 metastatic PTC among TFIs in cervical LNs.” should be transferred into the Discussion section.
FIGURE LEGENDS
Legend 2: Legend have information already present in the manuscript that should be removed; as an example, the sentence “Papillary thyroid carcinoma (PTC)-like nuclear features (NFs) were determined by following 18 the scoring system in a previous paper (9); scores 0 and 1 were considered absence of PTC-like NFs, 19 score 2 was considered questionable PTC-like NFs, and score 3 was considered presence of PTC- 20 like NFs.”, should be removed from the legend, as it was already mentioned on the text.
Regarding the other figures (figures 3 to 5), I´m not sure if have a legend or if the text below the images belongs to the result section. Please clarify.
REFERENCES
Reference 1 is not correct; please amend.
Some journals are written in full (as Histopathology) and others abbreviated (as Sci Rep); please uniform the criteria.
Author Response
MAJOR COMPULSORY CHANGES
The manuscript should be reviewed by an English native person, as there are multiple spelling lapses.
Response: Sorry for about this, but this manuscript has already proofread the English language by the Editage (https://www.editage.jp). Revised version has been double checked by the Editage.
MATERIAL AND METHODS
There is not a clear cut between “material and methods” and “results”. In fact, table 1 (Summary of the results) is mentioned in material and methods section…
Response: Table 1 is now mentioned in results section. Title of table 1 has changed as “Summary of clinicopathological profiles of the patients and results in this study”.
2.1. Sample collections- Authors should have a control group of people with ETT devoid thyroid tumours (for example ETT collected from cadaver’s devoid thyroid tumors).
Response: In our database including autopsy, any cases of ETT in LNs devoid thyroid tumours were not found, because LN resection were not carried out in patient devoid thyroid tumours. Therefore, this study used two cases of WDT-UMP and FA with ETT as a control. Additionally, we analyzed 7 cases of accessory thyroid tissue resected from patients with thyroid cancers, and found that they were negative for mBRAF, HBME-1, and galectin-3, and 53BP1 immunoreactivities.
2.3. IHC for mBRAF, Hector Battifora mesothelial epitope (HBME)-1, and galectin-3 expression- Please insert information regarding positive and negative controls used on IF and in IHQ.
Response: We inserted information about positive and negative controls for IHC in the 2.3 section.
DISCUSSION SECTION
Lines 160-161 - Please clarify the sentence, as it is difficult to understand “This suggested not true metastasis; however, it suggested ETT in LNs and in LNs from cases of WDT-UMP and FA.” as there are some verbs missing.
Response: Thank you for indication about this. We agree with your suggestion. It was corrected as following, “This suggested that TFI without PTC-like NF were not true metastasis but ETT in LNs as well as in LNs from cases of WDT-UMP and FA.”
Lines 169-170 - Please clarify the sentence: “IF increases with higher malignant potential in FFPE tissues of thyroid follicular tumors, such as FA, minimally invasive follicular carcinoma (FC), and widely invasive FC (5).” as FA (follicular adenoma) has not malignant potential. I suggest authors to replace “malignant potential” to “biological aggressiveness”.
Response: OK, “malignant potential” has been replaced by “biological aggressiveness” as following this suggestion.
REFERENCES
There are very few references (n=12), and most of the references cited (7/12) have one or more authors from the present manuscript. More references should be insert in the manuscript.
Response: More references inserted in the manuscript.
MINOR CHANGES
The text has different font letters; font must be uniformed.
INTRODUCTION
Line 9- Amend the sentence “of autopsy cases are without thyroid cancer” to “of autopsy cases without thyroid cancer”
Response: This was accordingly corrected.
MATERIALS AND METHODS SECTION
2.1. Sample collections lines 4-5 - Please replace “…following the scoring system in a previous paper…” to “…following a scoring system previously published…” or …following the scoring system described by Nikiforov et al. (2016).”
Response: This was accordingly corrected.
2.1. Sample collections lines 24-25- Before this sentence “…Representative pathological images of TFI in this study are shown in Figure 2….” there should be a paragraph. And the previous sentence should be insert after the reference to “formalin-fixed, paraffin embedded tissues”, as classification is always performed after the routine histologic procedure. I suggest the following sequence: “All available samples were formalin-fixed and paraffin-embedded …. Representative pathological images… Clinicopathological profiles of the patients are summarized in Table 1.”
Response: This was accordingly corrected.
2.2. IF analysis for 53BP1 expression, line 7- Please replace “…using a high standard all-in-one fluorescence microscope…” to “…using a high standard fluorescence microscope…
Response: This was accordingly corrected.
2.2. IF analysis for 53BP1 expression, line 10- Please replace “…nuclear foci:1” for “…nuclear foci: 1”.
Response: This was accordingly corrected.
RESULT SECTION
3.1., lines 75-76- Please remove from the title “expression using IF and immunoreactivity of” as it was already mentioned in the manuscript
Response: This was accordingly corrected.
3.2., lines 120-121- Please remove from the title “using IF and immunoreactivity of” as it was already mentioned in the manuscript
Response: This was accordingly corrected.
3.2., lines 128-130- The sentence “These data indicated the presence of benign 128 ETT in cervical LNs resected from patients with PTC, which should be differentiated from 129 metastatic PTC among TFIs in cervical LNs.” should be transferred into the Discussion section.
Response: This sentence was simply removed from 3.2. section, because similar sentence has already found in the Discussion section.
FIGURE LEGENDS
Legend 2: Legend have information already present in the manuscript that should be removed; as an example, the sentence “Papillary thyroid carcinoma (PTC)-like nuclear features (NFs) were determined by following 18 the scoring system in a previous paper (9); scores 0 and 1 were considered absence of PTC-like NFs, 19 score 2 was considered questionable PTC-like NFs, and score 3 was considered presence of PTC- 20 like NFs.”, should be removed from the legend, as it was already mentioned on the text.
Regarding the other figures (figures 3 to 5), I´m not sure if have a legend or if the text below the images belongs to the result section. Please clarify.
Response: Legend 2 was accordingly corrected. Legends of figures 3 and 4 have been also shorten. Legend 5 has not been modified to detail findings.
REFERENCES
Reference 1 is not correct; please amend.
Some journals are written in full (as Histopathology) and others abbreviated (as Sci Rep); please uniform the criteria.
Response: References corrected.
Reviewer 3 Report
The current research article “A New Indicator to Differentiate Thyroid Follicular Inclusions in Cervical Lymph Nodes from Patients with Thyroid Cancer’ by Otsubo et al reports 53BP1 as a new biomarker to differentiate thyroid follicular inclusions (TFIs) in cervical lymph nodes (LN) by comparing the mutational analyses of primary thyroid cancers. Using H&E staining, immunohistochemistry and immunofluorescence techniques, the authors have demonstrated differences in 53BP1 expression in TFIs without papillary thyroid carcinoma (PTC) like nuclear features (NF) and with questionable PTC-like NF or metastatic PTC. While the article is well written and addresses the hypothesis appropriately with methods and results, following are some minor suggestions to be made to further improve the quality of this article:
1. Please include a scale bar for the microscopic images in the figure or include the exact magnification in the figure legend.
2. Figures 3, 4 and 5 are missing figure legends. While most information has been included in the following paragraphs, it would be appropriate to add figure legends with each figure in the article.
3. Please include arrows/pointers in the figure wherever possible to show specific details (e.g., nuclear foci) and add information in the figure legends.
4. Immunofluorescence images aren’t clear and appear to be blurry, especially figure 5, please replace with better images.
5. Please add some more information on 53BP1 -> its activity; using as a target in cancer; future implications etc., in the introduction or discussion to make the article more accessible to a broader audience.
Author Response
- Please include a scale bar for the microscopic images in the figure or include the exact magnification in the figure legend.
Response: We added a scale bar for the microscopic images in the figure.
- Figures 3, 4 and 5 are missing figure legends. While most information has been included in the following paragraphs, it would be appropriate to add figure legends with each figure in the article.
Response: Each figure legend has been added to appropriate parts.
- Please include arrows/pointers in the figure wherever possible to show specific details (e.g., nuclear foci) and add information in the figure legends.
Response: We added arrowheads and arrows to indicate type of 53BP1 expression in each IF photos.
- Immunofluorescence images aren’t clear and appear to be blurry, especially figure 5, please replace with better images.
Response: We have replaced figures with higher resolution images, and supplementary provided two photos of IF images from Figures 3 and 4 at larger size.
- Please add some more information on 53BP1 -> its activity; using as a target in cancer; future implications etc., in the introduction or discussion to make the article more accessible to a broader audience.
Response: We have added more information on role of 53BP1 activity in human tumors in the 4th paragraph of the Discussion section.
Round 2
Reviewer 2 Report
Dear authors
This version is clearly improved regarding the previous version. Congratulations.